# Waterproof and Moisture-Permeable Polyurethane Nanofiber Membrane with High Strength, Launderability, and Durable Antimicrobial Properties

**DOI:** 10.3390/nano12111813

**Published:** 2022-05-25

**Authors:** Yong Xia, Lifen He, Jundan Feng, Sijun Xu, Lirong Yao, Gangwei Pan

**Affiliations:** 1National & Local Joint Engineering Research Center of Technical Fiber Composites for Safety and Protection, Nantong University, Nantong 226019, China; XiaYong1124@hotmail.com (Y.X.); Feng5017@hotmail.com (J.F.); pangangwei@ntu.edu.cn (G.P.); 2Jiangsu Jicui Advanced Fiber Material Research Institute Co., Ltd., Nantong 226010, China; helifen@jgafm.com

**Keywords:** silver nanoparticles, electrospinning, antibacterial, washing resistance

## Abstract

Nanofiber membrane has high biological protection function because of its good waterproof and moisture permeability properties. However, this membrane usually lacks active antimicrobial properties, limiting the application in reusable bioprotective textiles. Herein, waterborne polyurethane-capped Ag nanoparticles (AgNPs) were synthesized by reducing silver nitrate in water by sodium borohydride in the presence of polyurethane. AgNP-embedded thermoplastic urethane (TPU) nanofiber membrane was prepared by electrospinning a mixed solution of AgNPs and TPU. As-prepared membranes with Ag content of 50–300 mg·kg^−1^ have an average diameter of 0.75, 0.64, and 0.63 μm and good fiber uniformity. The doping of AgNP-embedded nanomembrane showed increased breaking force probably because of the induced crystallization effect. Test results showed that as-prepared TPU nanofiber membrane with silver content as low as 100 mg·kg^−1^ showed good washing resistance. The antibacterial rates of *E. coli* and *S. aureus* remained 99.99% with 50 times of soaping or chlorine washing. The corresponding waterproof and moisture permeability properties of nanofiber membrane with a thickness of 0.1 mm remained nearly unchanged, i.e., moisture permeability of around 2600 g·m^−2^ per 24 h and the hydrostatic pressure resistance of around 400 Pa after 50 times of soaping or chlorine washing.

## 1. Introduction

Nowadays, epidemics occur frequently all over the world, and medical protective clothing as emergency protective materials plays an important role in dealing with the spread of pathogen especially that the demand for reusable protective clothing increases sharply. Bioprotective clothing comes into contact with air, liquid, and dust during use, but bacteria cannot exist independently and adhere to aerogel, liquid, and dust to spread [1,2]. The basic demand for bioprotective textiles is high bioprotection, which can be achieved by improving barrier properties and antimicrobial treatment.

Nano inorganic antibacterial agents have the advantages of broad antibacterial spectrum, long effective antibacterial period, low toxicity, no drug resistance, and high safety. As a kind of nano-inorganic antibacterial agent, nanosilver has been widely used in environmental purification, medical treatment, cosmetics, textiles, and other fields. In the textile industry, nanosilver can give fabrics antibacterial, antistatic, antielectromagnetic radiation properties and other functions [3,4,5]. Its antibacterial mechanism is that nanosilver binds to the cell wall or cell membrane of the pathogen and directly enters the thallus and quickly binds to the sulfhydryl group of oxygen metabolism, thereby blocking metabolism and making it lose its activity until death [6,7,8]. However, the nanosilver antibacterial agent endows fabrics with high antibacterial activity and has the disadvantages of easy falling off from fabrics and being easily inactivated by oxidation, limiting its wide application [9,10,11]. Therefore, how to protect nanosilver antibacterial agents effectively has become a research hotspot [12,13,14]. Xu et al. [15] used α-ketoglutarate to modify chitosan and synthesized nanosilver and chitosan derivative adhesives for the functional finishing of textiles. Results show that after 30 times of continuous washing of cotton fabric with antibacterial coating, the content of silver nanoparticles (AgNPs) on the surface of cotton fabric decreases to 37.6%, and the antibacterial rates of *Staphylococcus aureus* and *Escherichia coli* are 95% and 96%, respectively.

With the advancement of technology, bioprotective textiles began to develop towards high comfort and reusability. Reusable medical protective fabrics require comprehensive properties, such as good barrier performance of micro- and nanoparticles, good moisture permeability, active microbial resistance, and excellent washing resistance. Among numerous filter materials, polarized nanofiber membrane consisting of numerous nanogaps is an ideal candidate material because it can electrostatically adsorb and isolate micro- and nanoparticles and allow moisture to pass through. By further integrating antimicrobial AgNPs, the nanofiber membrane is further endowed with reusable bioprotective properties [16,17]. However, the precondition is that AgNPs should possess good chemical stability and excellent binding strength with nanofiber materials to ensure durable antibacterial activities and good washing fastness. Compared with coating AgNPs on the surface of nanofibers, AgNP-embedded nanofibers show high chemical stability and washing resistance. However, this should cap a layer of appropriate polymer compatible with the electrospinning solution on the surface of AgNPs and synthesize core–shell structure AgNPs to protect AgNPs from agglomeration and oxidation [18,19,20].

In this paper, water-soluble waterborne polyurethane (WPU)-capped AgNPs are prepared through the reduction in silver nitrate precursor by sodium borohydride with WPU as a protecting agent. The WPU macromolecule contains amide groups that can complex with AgNPs and form a stable protecting layer, imparting NPs with good physical and chemical stability. To prepare a durable antibacterial nanofiber nonwoven fabric, as-prepared AgNPs are directly mixed into the TPU solution, and the porous and polarized AgNP-embedded TPU nanofiber membrane is obtained by electrostatic spinning. Given that AgNPs are protected by WPU and embedded inside the nanofiber, the release rate of silver ions is slow, thereby achieving a long-term antibacterial effect. Furthermore, as-prepared nanofiber membrane can withstand 50 washes with silver content as low as 100 mg·kg^−1^, and the corresponding antibacterial performance can still maintain 99.99%, suggesting excellent reusable performance.

## 2. Materials and Methods

### 2.1. Chemicals

TPU particles were obtained from the Shanghai Yuanneng Plastic Co., Ltd. (Shanghai China). AgNO_3_ (>99.8%), ammonia (25–28%), sodium borohydride (≥97%), sodium chloride (99.5%), monosodium phosphate (99%), WPU (32%), DMF (≥99.5%), and acetone (≥99.5%) were obtained from Sinopharm Chemical Reagent Co., Ltd. (Shanghai, China). Gram-negative *E. coli* (ATCC 25922) and Gram-positive *S. aureus* (CMCC 26003) were obtained from the Shanghai Luwei Technology Co., Ltd. (Shanghai, China). Nutrient agar and broth were obtained from the Hangzhou Baisi Biology Technology Co., Ltd. (Hangzhou, China).

### 2.2. Preparation and Characterization of WPU-Capped AgNPs

About 0.78 g silver nitrate was added to 100 mL deionized water to prepare a 7.8 g·L^−1^ silver nitrate solution. About 0.1 g sodium borohydride was added into 100 mL deionized water to prepare 1 g·L^−1^ sodium borohydride solution. The silver nitrate solution was mixed with an appropriate amount of ammonia water to prepare the silver ammonia solution. About 40 mL silver ammonia solution was added with 4 g WPU (solid content = 32%), 3 mL sodium borohydride solution to the above mixed solution, and deionized water to obtain a 50 mL volume. About 4000 mg·L^−1^ WTU-capped AgNPs solution was prepared by the in situ chemical reduction method and stored in a brown glass bottle [21,22,23,24,25].

WTU-capped AgNPs were characterized by transmission electron microscopy (TEM; JEM-2100, JEOL, Tokyo, Japan), a particle analyzer (90 plus Zeta, Brookhaven instruments corporation, Holtsville, NY, USA) and ultraviolet spectrophotometry (UV-vis; TU-1901, Beijing Puxi General Instrument Co., Ltd., Beijing, China).

### 2.3. Preparation and Characterization of AgNP-Embedded TPU Nanofiber Membrane

About 5 g TPU particles were added into a 20 g mixture of acetone and DMF (mass ratio 1:1), and the mixture was heated in a water bath maintained at 60 °C and stirred by magnetic force until complete dissolution, thus obtaining a 20% TPU solution. The same method was used to prepare five parts. Different amounts of 4000 mg·L^−1^ WTU-capped AgNP solution were added into TPU solution, thus obtaining mass ratios of AgNPs to TPU of 0, 50, 100, 200, and 300 mg·kg^−1^. Mechanical stirring dispersed WTU-capped AgNPs evenly in the TPU system, and the TPU spinning solution with different AgNP contents was obtained. AgNP-embedded TPU nanofiber membrane with AgNP contents of 0, 50, 100, 200, and 300 mg·kg^−1^ were prepared by electrospinning. Experimental parameters were as follows: spinning solution flow rate, 3 mL·h^−1^; injection volume, 3 mL; anode voltage, 15 kV; acceptance distance, 15 cm; and rotation speed of receiving roller, 100 r·min^−1^. The prepared AgNP-embedded TPU nanofiber membrane was baked in an oven at 60 °C for 6 h to volatilize the residual solvent [26,27,28].

The viscosity, conductivity, and surface tension of mixture of AgNPs and TPU were measured by a rotary rheometer (Antonpa Co., Ltd., Graz, Austria), a conductivity meter (Changhong precision instrument Co., Ltd., Mianyang, China), and a surface tension meter (Yide precision instrument Co., Ltd., Guangzhou, China). The morphology of samples was observed by scanning electron microscopy (SEM; ZEISS Gemini SEM300, Carl Zeiss Company, Oberkochen, Germany). The elements content was analyzed by energy dispersive spectroscopy (EDS) mapping. The chemical structure was analyzed by Fourier transform infrared spectroscopy (FTIR; Thermo Nicolet iS50, Thermo Fisher Scientific shier science & technology company, Waltham, MA, USA). The tensile strength of as-prepared nanofiber membrane was tested by a universal material testing machine (INSTRON 1346, Instrang (Shanghai) Test Equipment Trading Co., Ltd., Shanghai, China).

### 2.4. Release Kinetics of Silver Ions

About 5 g NaCl and 2.2 g NaH_2_PO_4_·2H_2_O were added into 1 L deionized water, and the pH of the solution was adjusted to pH 5.5 by using 0.01 mol·L^−1^ HCl and 0.01 mol·L^−1^ NaOH to prepare artificial sweat.

AgNP-embedded TPU nanofiber membrane could achieve antibacterial effect by releasing silver ions. We explored the silver ion release performance of AgNP-embedded TPU nanofiber membrane with different AgNP contents (50, 100, and 300 mg·kg^−1^) in artificial sweat. About 1 g AgNP-embedded TPU nanofiber membrane was immersed in artificial sweat, and the bath ratio was 1:50. At immersion times of 5 min, 1 h, 3 h, 6 h, 10 h, and 24 h, 2 mL immersion liquid was collected, and the silver ion content in the immersion liquid was measured using inductively coupled plasma-mass-spectrometry (ICP-MS).

In addition, we explored the release performance of silver ions from AgNP-embedded TPU nanofiber membrane at different pH values. About 1 g of 100 mg·kg^−1^ AgNP-embedded TPU nanofiber membrane was dipped in artificial sweat, and the pH of the dipping solution was adjusted to pH 8.5 by using 0.01 mol·L^−1^ HCl and 0.01 mol NaOH, and the bath ratio was 1:50. At dipping times of 5 min, 1 h, 3 h, 6 h, 10 h, and 24 h, 2 mL dipping solution was collected, and the silver ion content in the dipping solution was measured using ICP-MS [29,30,31].

### 2.5. Antibacterial Tests

The antibacterial property of AgNP-embedded TPU nanofiber membrane was tested by improved oscillation method (GB/T 20944.3-2008, China). Gram-negative *E. coli* and Gram-positive *S. aureus* were selected as test strains. Bacteria were shaken in nutrient broth at 37 °C and 130 rpm for 18–20 h and diluted into 105 colony-forming units (CFU)·mL^−1^ in 4.2 g·L^−1^ phosphate buffer (PBS). About 0.1 g AgNP-embedded TPU nanofiber membrane was placed into 5 mL centrifuge tubes and added with 3 mL diluted bacterial solution. Then, the mixture was shaken at 37 °C for 18–24 h at 130 rpm. After the nutrient agar was poured into a culture dish, cooled, and solidified, the bacterial solution in the centrifuge tube containing the sample was diluted 10, 10^2^, and 10^3^ times by using 4.2 g·L^−1^ PBS. The undiluted and 10, 10^2^ and 10^3^ fold diluted bacterial solutions were added into four areas of the culture dish and cultured at 37 °C for 18–24 h. Pure TPU nanofiber membrane was used as control sample. The bacteriostatic rate was calculated as follows:(1)$$\mathrm{Antibacterial}\;\mathrm{rate}=\frac{B-A}{B}\;\times \;100\%,$$
where A is the number of colonies in any quarter of the test culture dish, and B is the number of colonies in the same quarter of the control culture dish [32,33].

### 2.6. Waterproof and Moisture Permeability Test

The cup test method (GB/T 1037-1988, China) was used to determine the moisture permeability of AgNP-embedded TPU nanofiber membrane with 0.1 mm AgNP contents of 0, 50, 100, 200, and 300 mg·kg^−1^.

The hydrostatic pressure method (GB/T 4744-1997, China) was used to test the waterproof performance of AgNP-embedded TPU nanofiber membrane with 0.1 mm AgNP contents of 0, 50, 100, 200, and 300 mg·kg^−1^.

### 2.7. Wash Resistance Test

The soaping fastness method (GB/T 3921-2008, China) was used to test the washing resistance of AgNP-embedded TPU nanofiber membrane with AgNP contents of 50, 100, 200, and 300 mg·kg^−1^.

The color fastness to chlorinated water (GB/T 8433-2013, China) was used to test the chlorine bleaching resistance of AgNP-embedded TPU nanofiber membranes with AgNP contents of 50, 100, 200, and 300 mg·kg^−1^.

## 3. Results and Discussion

### 3.1. Preparation and Characterization of WPU-Capped AgNPs

WPU has hydrophilic and hydrophobic segments and can form stable micelles in water. During the reaction, silver ions adsorbed by amide groups were quickly reduced by sodium borohydride and subsequently formed a stable complex structure because of the Ag-N interaction. As a result, AgNPs gained small particle size and good solution dispersion and by capping negatively charged WPU. As shown in Figure 1a,b, WPU-capped AgNPs had a small particle size range of 10–20 nm (Figure 1b). The typical adsorption peak of AgNPs at 434 nm in the UV–Visible (vis) spectra confirmed the formation of AgNPs (Figure 1c). As shown in Figure 1g,h, when the temperature was raised to 120 °C, the color and absorbance intensity of the UV–vis spectra of the WPU-capped AgNPs remained unchanged because AgNPs were encapsulated in WPU and formed a core–shell nanostructure, thereby imparting good physical and chemical stability. This result indicated low Ag^+^ release, ensuring long-term antibacterial ability. In addition, WPU and TPU showed similar structure. Therefore, WPU-capped AgNPs were compatible with TPU. As shown in Figure 1i, the surface potential of AgNPs was approximately −53.4 mV.

### 3.2. Preparation and Characterization of AgNP-Embedded TPU Nanofiber Membrane

As shown in Figure 1e, the addition of AgNPs increased the viscosity of spinning solution because the movement of TPU molecular chain was restricted by forming intermolecular hydrogen bonds between TPU macromolecules and WPU after adding WPU-capped AgNPs into TPU solution, thereby increasing the viscosity of spinning solution. With increasing Ag content, the conductivity of spinning solution also increased because AgNPs increased the number of ions in the spinning liquid system by releasing Ag^+^ (Figure 1f). Similarly, AgNPs increased the surface tension of spinning solution (Figure 1f). To some extent, the spinning performance of TPU solution could be improved by increasing the viscosity and conductivity of spinning solution and reducing the surface tension.

Figure 2a–c indicate that the nanofiber membranes made from spinning solutions with Ag contents of 50, 100, and 300 mg·kg^−1^ through electrostatic spinning process had good fiber uniformity and spinnability. In addition, the surface of pure TPU nanofibers was smoother than that of 100 mg·kg^−1^ of AgNP-embedded nanomembrane, in which a small number of AgNPs was found to attach to the surface of TPU nanofibers (Figure 2d–i). The EDS mapping analysis showed that AgNPs were uniformly dispersed in TPU nanofibers (Figure 3a–e).

The FTIR spectra (Figure 4a) indicated that the pure TPU nanofiber membrane had an N-H stretching vibration peak at 3330 cm^−1^, a C-H stretching vibration peak at 2960 cm^−1^, a -CH_2_ symmetrical vibration peak at 2870 cm^−1^, a free C=O stretching vibration peak at 1730 cm^−1^, and a characteristic peak of hydrogen bonding carbonyl stretching vibration at 1700 cm^−1^. Compared with those of pure TPU nanofiber membrane, the FTIR spectra of AgNP-embedded TPU nanofiber membrane showed no new absorption peak, indicating no new chemical bond between AgNPs and TPU matrix.

The strength of nanofiber membrane is one of the most important properties in its application in bioprotective fabrics. As shown in Figure 4b, the breaking force of AgNP-embedded TPU nanofiber membrane improved by increasing the Ag content compared with that of pure membrane. Similar to the concrete structure, AgNPs in TPU nanofibers showed strong induced crystallization effect for TPU linear macromolecules.

### 3.3. Release Kinetics of Silver Ions

Figure 4c shows the release kinetics of Ag^+^ from TPU nanofiber membrane with different Ag contents in artificial sweat. With increasing Ag content, the release of Ag^+^ also increased because with increased AgNP loading, the number of AgNPs participating in the dissociation reaction increased. Thus, the release concentration of Ag^+^ increased. In addition, the release amounts of Ag^+^ from 100 and 300 mg·kg^−1^ AgNP-embedded TPU nanofiber membranes were similar. This result indicated that simply increasing the Ag content could not increase the release of Ag^+^ indefinitely. When the membrane had AgNP content exceeding 100 mg·kg^−1^, Ag release basically did not increase compared with that with AgNP content of 50 mg·kg^−1^, indicating decreased Ag^+^ release efficiency.

Figure 4d shows the release kinetics of Ag^+^ from AgNP-embedded TPU nanofiber membrane immersed in artificial sweat under different pH conditions. The pH of the aqueous solution affected the Ag^+^ release performance of AgNP-embedded TPU nanofiber membrane. With increased pH, the release of Ag^+^ from AgNP-embedded TPU nanofiber membrane decreased evidently. At pH 5.5, the release amount of silver ions was much larger than that at pH 8.5. The reason was that the released Ag^+^ from AgNP-embedded TPU nanofiber membrane came from AgNPs in nanofibers. The chemical reaction of AgNPs releasing Ag^+^ was a synergistic oxidation reaction process. At first, under the joint action of O_2_ and H^+^, AgNPs underwent oxidation reaction to generate some intermediates of Ag^+^ and peroxide, which further reacted with AgNPs to generate Ag^+^ and H_2_O. Therefore, the release of Ag^+^ was positively correlated with the concentration of H^+^, and the release rate of Ag^+^ was remarkably increased at low pH.

In addition, all Ag^+^ release kinetics shown in Figure 4c,d showed two release peaks. The first weak release peak at 200 min was derived from Ag^+^ generated by AgNPs on the surface of TPU nanofibers. The second strong peak at 600 min came from Ag^+^ release of AgNPs inside nanofibers.

### 3.4. Antibacterial Tests

The addition of AgNPs endowed the TPU nanofiber membrane with excellent antibacterial properties. Excellent antibacterial effect could be achieved with a low load of AgNPs, which was because the prepared AgNPs had small particle size, high antibacterial activity, and good dispersibility. In addition, the AgNP-embedded TPU nanofiber membrane prepared by electrospinning had high specific surface area, which was conducive to the release of Ag^+^. AgNPs was directly doped into TPU solution and electrospun to prepare nanofiber composite membrane. Given that most AgNPs were embedded inside the TPU nanofiber, the nanofiber membrane exhibited good washing resistance. As shown in Table 1 and Figure 5, even at Ag content of 50 mg·kg^−1^, the antibacterial rates to *E. coli* and *S. aureus* reached 90.37% and 97.57%, respectively, after 50 times of washing. With Ag content increasing to 100 mg·kg^−1^, the antibacterial rates were over 99.99% after 50 times of washing, suggesting strong antibacterial performance and good washing resistance. Compared with traditional nano-silver functional fabrics, the as-prepared nanofiber membrane has the advantages of high antibacterial efficiency and good washing resistance because of the good compatibility of WPU-capped AgNPs with TPU matrix [34,35,36].

The AgNP-embedded TPU nanofiber membrane also exhibited excellent chlorine-washing resistance. The antibacterial rates of 50 mg·kg^−1^ AgNP-embedded TPU nanofiber membrane against *E. coli* and *S. aureus* reached 99.99% and 99.99%, respectively, after 50 times of chlorine washing (Table 2 and Figure 6). This finding was because the core–shell-structured WPU-capped AgNPs were encapsulated in TPU nanofibers, thus effectively isolating NPs from the external environment.

### 3.5. Waterproof and Moisture Permeability Test

Given the numerous nano-sized voids formed by nanofibers, the as-prepared nanomembrane could effectively block liquid and dust and allow water vapor to pass through. As shown in Figure 7a, with increasing AgNPs from 0 mg·kg^−1^ to 300 mg·kg^−1^, the moisture permeability and waterproof performance of AgNP-embedded TPU nanofiber membrane remained unchanged. In addition, 100 mg·kg^−1^ AgNP-embedded TPU nanofiber membrane could tolerate 0–50 times of soaping or chlorine washing (Figure 7a–c), indicating good moisture permeability and waterproof performances.

## 4. Conclusions

WPU-capped AgNPs with particle sizes of 10–20 nm were successfully prepared by the in situ chemical reduction in AgNO_3_ by sodium borohydride in WPU solution. As-prepared AgNPs showed good solution stability and high temperature resistance. The electrospun nanofiber membranes with Ag content of 50–300 mg·kg^−1^ had average diameters of 0.75, 0.64, and 0.63 μm and showed good fiber uniformity. The doping of AgNPs in nanofibers increased the breaking force because of the induced crystallization effect. The nanofiber membrane showed two Ag^+^ release peaks in artificial sweat, i.e., 60–80 and 120–140 µg·L^−1^, which were derived from AgNPs on the surfaces and inside of nanofibers, respectively. The as-prepared nanofiber membrane showed good washing resistance. It maintained high antibacterial activities against *E. coli* and *S. aureus* after 50 times of soaping or chlorine washing with antibacterial rates over 99.99% and 99.99%, respectively, when Ag content was over 100 mg·kg^−1^. In addition, the waterproof and moisture permeability properties of 0–300 mg·kg^−1^ AgNP-embedded nanofiber membrane with a thickness of 0.1 mm remained nearly unchanged with 0–50 times of soaping or chlorine washing. The corresponding moisture permeability maintained around 2600 g·m^−2^ per 24 h and the hydrostatic pressure resistance was around 400 Pa with 0–50 times of soaping or chlorine washing. In summary, the porous AgNP-embedded TPU nanofiber membrane possessed high antibacterial activities, excellent washing resistance and good oxidation resistance, and had potential as an antibacterial membrane material for the preparation of reusable bioprotective laminated composite clothing.

## Figures and Tables

**Figure 1 nanomaterials-12-01813-f001:**
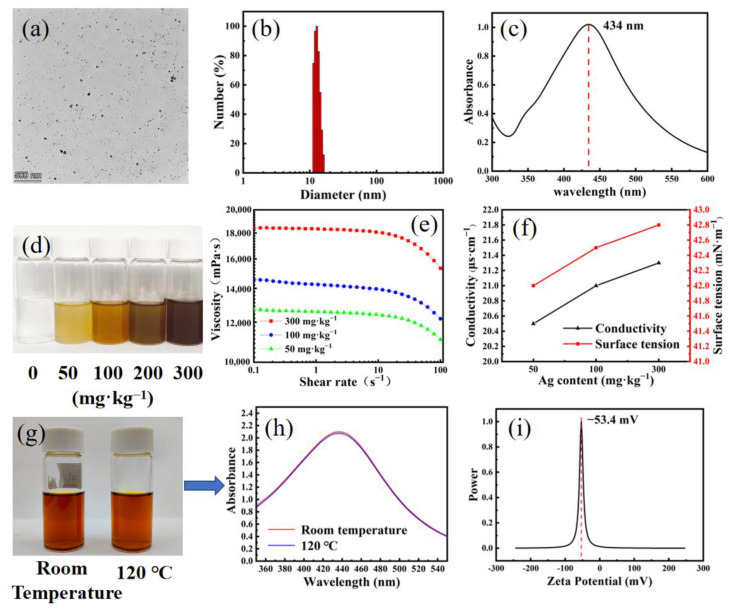
(**a**) TEM image, (**b**) particle size distribution, and (**c**) UV–vis spectrum of AgNP solution. (**d**) Photos of TPU solutions with gradient Ag content. (**e**) Viscosity of TPU solutions. (**f**) Conductivity and surface tension of TPU solutions with Ag content range of 50–300 mg·kg^−1^. (**g**) Images and (**h**) UV–vis spectra of WPU-capped AgNPs solutions (30 mg·L^−1^) after treatment at different temperatures for 2 h. (**i**) Zeta potential of WPU-capped AgNPs.

**Figure 2 nanomaterials-12-01813-f002:**
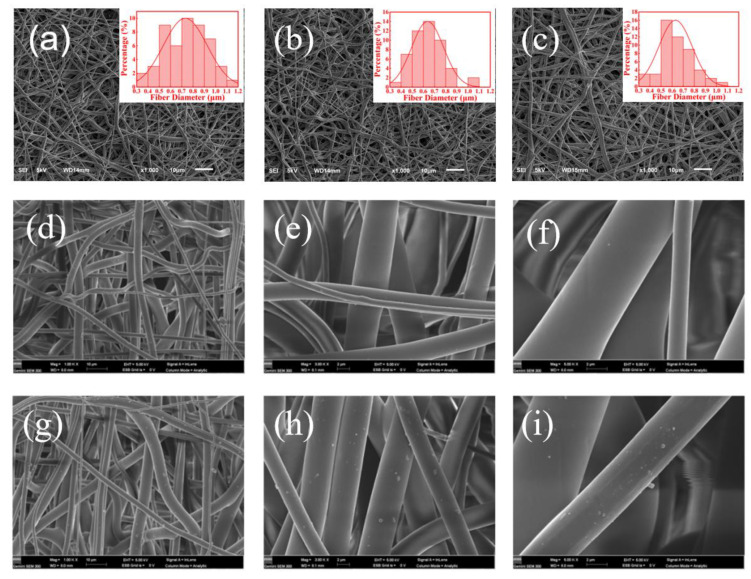
(**a**–**c**) SEM images of AgNP-embedded TPU nanofiber membrane with AgNP contents of 50, 100, and 300 mg·kg^−1^. SEM images of (**d**–**f**) pure TPU nanofiber membrane and (**g**–**i**) AgNP-embedded TPU nanofiber membrane with AgNP contents of 100 mg·kg^−1^.

**Figure 3 nanomaterials-12-01813-f003:**
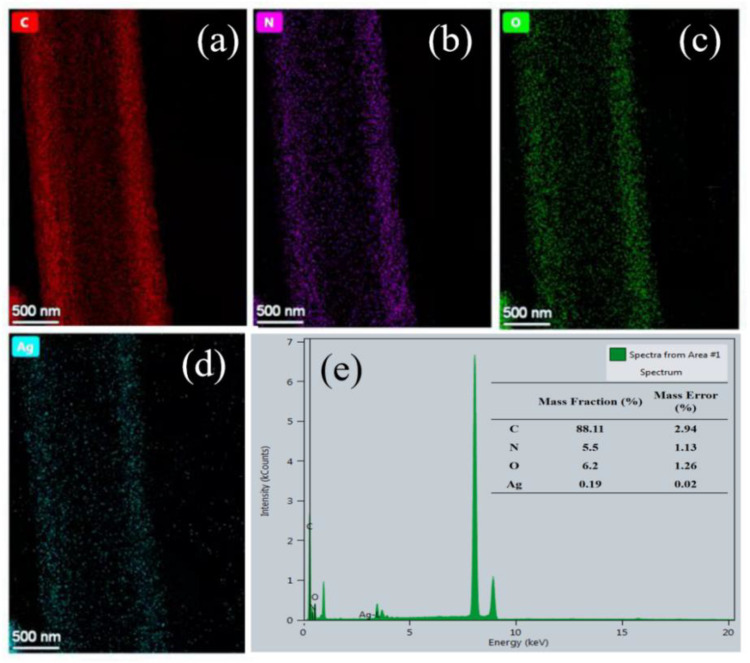
(**a**–**e**) EDS mapping diagram of AgNP-embedded TPU nanofiber membrane.

**Figure 4 nanomaterials-12-01813-f004:**
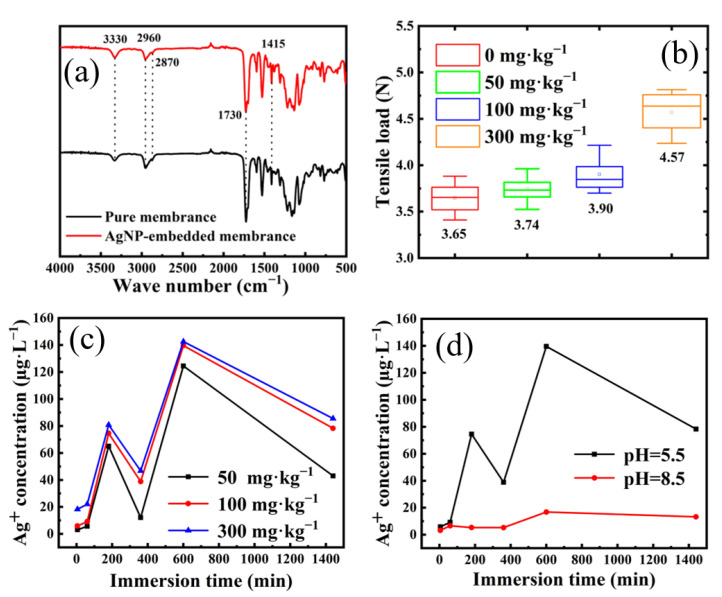
(**a**) FTIR spectra of pure and AgNP-embedded TPU nanomembrane. (**b**) Tensile load of TPU nanofiber membrane with different Ag contents. (**c**) Ag^+^ release kinetics of TPU nanofiber membrane with different Ag contents in artificial sweat. (**d**) Ag^+^ release kinetics 100 mg·kg^−1^ AgNP-embedded TPU nanofiber membrane at pH 5.5 and 8.5.

**Figure 5 nanomaterials-12-01813-f005:**
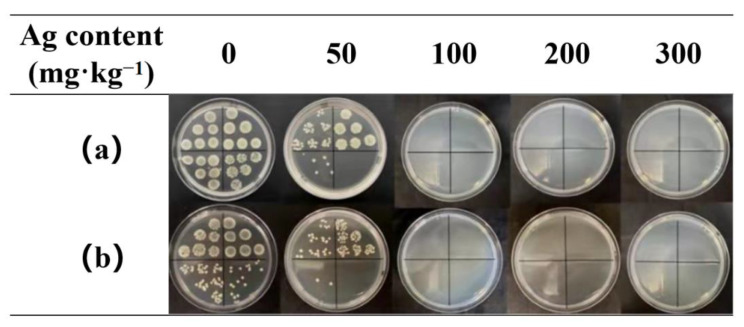
Petri dishes containing (**a**) *E. coli* and (**b**) *S. aureus* with TPU nanofiber membrane samples with Ag content ranging from 0 mg·kg^−1^ to 300 mg·kg^−1^ after 50 times of washing.

**Figure 6 nanomaterials-12-01813-f006:**
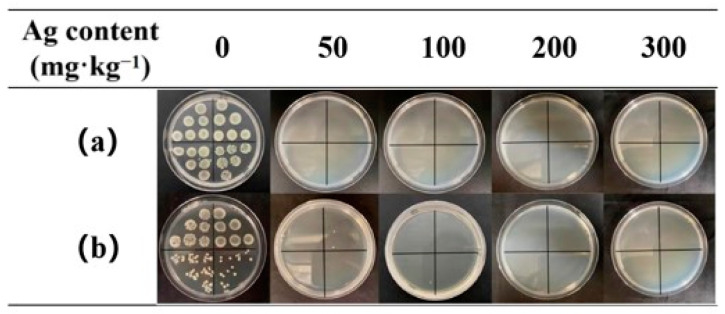
Petri dishes containing (**a**) *E. coli* and (**b**) *S. aureus* with TPU nanofiber membrane samples with Ag content ranging from 0 mg·kg^−1^ to 300 mg·kg^−1^ after 50 times of chlorine washing.

**Figure 7 nanomaterials-12-01813-f007:**
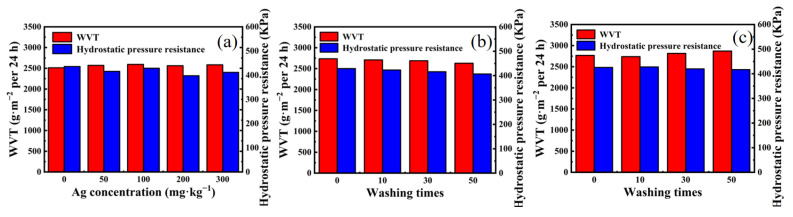
(**a**) Waterproof and moisture permeability properties of TPU nanofiber membrane with Ag content ranging from 0 mg·kg^−1^ to 300 mg·kg^−1^. Waterproof and moisture permeability properties of 100 mg·kg^−1^ AgNP-embedded TPU nanofiber membrane with 0–50 times of (**b**) soaping and (**c**) chlorine washing.

**Table 1 nanomaterials-12-01813-t001:** Antibacterial rates of TPU nanofiber membrane with Ag contents ranging from 0 mg·kg^−1^ to 300 mg·kg^−1^ after 50 times of standard washing.

Ag Content (mg·kg^−1^)	Antibacterial Rates (%)	
	*Escherichia coli*	*Staphylococcus aureus*
50	90.37	97.57
100	99.99	99.99
200	99.99	99.99
300	99.99	99.99

**Table 2 nanomaterials-12-01813-t002:** Antibacterial rates of TPU nanofiber membranes with Ag contents ranging from 0 mg·kg^−1^ to 300 mg·kg^−1^ after 50 times of chlorine washing.

Ag Content (mg·kg^−1^)	Antibacterial Rates (%)	
	*Escherichia coli*	*Staphylococcus aureus*
50	99.99	99.99
100	99.99	99.99
200	99.99	99.99
300	99.99	99.99

## Data Availability

The data that support the findings of this study are available from the corresponding author upon reasonable request.

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
