# Peer review of "Waterproof and Moisture-Permeable Polyurethane Nanofiber Membrane with High Strength, Launderability, and Durable Antimicrobial Properties"

_nanomaterials, 2022, doi:10.3390/nano12111813_

Round 1
Reviewer 1 Report
The work deals with the fabrication of electrospun membranes enriched with polyurethane-stabilized silver nanoparticles to be used as bioprotective fabrics. The topic is sure of relevant interest and the preliminary outcomes are promising. Overall, the manuscript is well-written, the Introduction is comprehensive, and the results are presented in a clear way and deeply discussed. Therefore, I am willing to suggest the manuscript publication in the present form.
Author Response
Thanks for your approval.
Reviewer 2 Report
My review
Waterproof and moisture-permeable polyurethane nanofiber membrane with high strength, launderability, and durable antimicrobial properties
In the last decade of the 20th century, polymer nanofibers gained great popularity in various branches of science and industry. The nanofibers create a mechanical barrier that prevents the entry of hazardous particles, without the use of hazardous chemicals.
Window with a nanofiber membrane, captures smog, exhaust fumes from car engines, pollen, dust and bacteria. However, the membrane usually does not have active antimicrobial properties that would limit its use in reusable bio-protective textiles.
Protective clothing used successfully in the work environment covers or replaces personal clothing to protect against hazards. Now that epidemics are frequent around the world, the demand for reusable protective clothing is increasing.
In the reviewed work, Waterproof and moisture-permeable polyurethane nanofiber membrane with high strength, launderability, and durable antimicrobial properties, water-based Ag nanoparticles coated with polyurethane - AgNP were synthesized by reducing silver nitrate in water by sodium borohydride (in the presence of polyurethane).
WTU capped AgNPs were characterized by transmission electron microscopy, a particle analyzer (90 plus Zeta, Brookhaven Instruments Corporation, USA) and ultraviolet spectrophotometry. The kinetics of silver ion release was investigated, and antibacterial tests were performed (the antibacterial index was determined).
Good antimicrobial activity could be achieved with low AgNPs loading. It should be noted that the membrane made of TPU nanofibers (thermoplastic polyurethane) with deposited AgNP - silver nanoparticles, prepared by the electrospinning method, had a high specific surface, which favored the release of Ag +.
To determine the humidity, the cup test method (GB / T 1037-1988, China) was used, the permeability of the TPU nanofiber membrane with AgNP deposited with AgNP content 0.1 mm 171 0, 50, 100, 200 and 300 mg · kg-1.
In the reviewed work, the results of the research / experiments conducted were included in appropriate, carefully prepared drawings, eg Figure 1. (a) TEM image, images and UV-Vis spectra of WPU-closed AgNPs solutions. In contrast, Figure 2 illustrates SEM images with AgNP embedded TPU nanofiber membrane, Figure 3 is the EDS mapping diagram of AgNP-embedded TPU nanofiber membrane, Figure 4 - FTIR spectra of pure and AgNP-embedded TPU nanomembrane.
All Ag + release kinetics shown in Figures 4 (c-d) showed two release peaks. The first weak release peak at 200 min was from Ag + generated by AgNPs on the surface of the TPU nanofibers. In contrast, the second strong peak at 600 min came from the silver nanoparticle (AgNPs) inside the nanofibers.
In order to implement the assumptions of the work, a test of water resistance and moisture permeability was carried out. Figure 7 shows the waterproofing and moisture permeability properties of a TPU nanofiber membrane.
The antibacterial coefficients of the TPU nanofiber membrane with Ag content are presented in the tables (Table 1.2). With the Ag content increasing to 100 mg · kg-1, the antimicrobial coefficients were about 100% after 50 times washing, suggesting a strong antibacterial effect and good resistance to washing.
As a result of the conducted research, it was found that the TPU nanofiber membrane embedded in AgNP had excellent antibacterial properties, resistance to washing and has the potential as an antibacterial membrane material for the preparation of reusable bio-protective laminated composite clothing.
The work is written correctly, the results of the research have been documented with tables and appropriate drawings, good scientific discussion. The thematic literature review carefully documents the scientific achievements to date.
Summing up, the reviewed work Waterproof and moisture-permeable polyurethane nanofiber membrane with high strength, launderability, and durable antimicrobial properties presents important practical research on high-strength polyurethane membranes, the use of which is very wide. These membranes, among other things, are also useful in the renovation of flat roofs covered, for example, with a PVC membrane.
In the opinion of the reviewer, the manuscript Waterproof and moisture-permeable polyurethane nanofiber membrane with high strength, launderability, and durable antimicrobial properties is suitable for printing after the linguistic correction of the text and careful emphasis on the essence of the research on polyurethane nanofiber membranes.
Author Response
Thanks for your approval.

Reviewer 3 Report
The manuscript "Waterproof and moisture-permeable polyurethane nanofiber membrane with high strength, launderability, and durable antimicrobial properties" deals with the production of polyurethane membranes loaded with silver nanoparticles (AgNPs) by electrospinning with the aim of obtaining a device with antimicrobial properties.
The work is well written and organized. Several analyses were performed on the polymeric membranes, obtaining intriguing results. Therefore, the publication is recommended; but after some revisions, as follows:
- Intrduction. The state of the art related on the production of membranes and porous materials loaded with AgNPs can be enlarged. For this purpose, see for instance these works: Baldino et al., , 12(3), 388; etc..
- Use "mL" instead of "ml"; etc..
- M&M. Explain the selection of the operating parametrs for electrospinning.
- R&D. Compare the results with the previous literature to underline the relevance of the present findings.
- Conclusions are a summary of the work. Rewrite in a more critical way.
- Correct typos.
